# Simultaneous Determination of Orelabrutinib, Zanubrutinib, Ibrutinib and Its Active Metabolite in Human Plasma Using LC-MS/MS

**DOI:** 10.3390/molecules28031205

**Published:** 2023-01-26

**Authors:** Lu-Ning Sun, Yang Zhao, Zhou-Yi Qian, Xiang-Long Chen, Hong Ma, Yu-Jiao Guo, Hao Shen, Yong-Qing Wang

**Affiliations:** 1Research Division of Clinical Pharmacology, First Affiliated Hospital of Nanjing Medical University, Nanjing 210029, China; 2School of Pharmacy, Nanjing Medical University, Nanjing 211166, China

**Keywords:** Bruton’s tyrosine kinase inhibitor, human plasma, LC-MS/MS

## Abstract

Ibrutinib, orelabrutinib, and zanubrutinib are all Bruton’s tyrosine kinase inhibitors, which have greatly improved the treatment of B-cell malignancies. In this study, an LC-MS/MS method was developed and validated for the determination of orelabrutinib, zanubrutinib, ibrutinib, and its active metabolite dihydrodiol ibrutinib in human plasma. The Ibrutinib-d5 was used as the internal standard. Pretreatment was performed using a simple protein precipitation step using acetonitrile. The ACQUITY UPLC HSS T3 column (2.1×50 mm, 1.8 μm) was used to separate the analytes, and the run time was 6.5 min. The mobile phase consisted of acetonitrile and 10 mM of ammonium formate, which contained 0.1% formic acid. The multiple reactions’ monitoring transitions were selected at m/z 428.1→411.2, 472.2→455.2, 441.1→304.2, 475.2→304.2 and 446.2→309.2 respectively for orelabrutinib, zanubrutinib, ibrutinib, dihydrodiol ibrutinib and ibrutinib-d5 using positive ion electrospray ionization. The standard curves were linear, from 0.400 to 200 ng/mL for ibrutinib and dihydrodiol ibrutinib, 1.00–500 ng/mL for orelabrutinib, and 2.00–1000 ng/mL for zanubrutinib. Selectivity, the lower limit of quantitation, precision, accuracy, matrix effect, recovery, stability, and dilution integrity all met the acceptance criteria of FDA guidance. This method was used to quantify the plasma levels of orelabrutinib, zanubrutinib, ibrutinib, and dihydrodiol ibrutinib in clinical patients.

## 1. Introduction

Bruton’s tyrosine kinase (BTK) is a pivotal signal protein in the B-cell receptor (BCR) pathway that is critical for the proliferation and survival of B-cells [1]. Targeting the BCR signaling pathway using the BTK inhibitors can lead to the downstream mitigation of cell growth, proliferation, adhesion, migration, and survival of malignant B cells. The appearance of the BTK inhibitors has dramatically improved the treatment of several B-cell malignancies, such as marginal zone lymphoma, Waldenström macroglobulinemia, chronic lymphocytic leukemia/small lymphocytic lymphoma (CLL/SLL), and mantle cell lymphoma (MCL) [2,3,4]. Ibrutinib (IBR) was the first BTK inhibitor approved by the FDA in 2013 as a breakthrough treatment for patients with CLL [4]. It is the best-studied BTK inhibitor, with the advantages of more accessible long-term information and more clinical experience [2]. However, its off-target effect and drug resistance led to the development of next-generation BTK inhibitors [5,6]. Zanubrutinib (ZAN) and orelabrutinib (ORE) are both second-generation BTK inhibitors with higher selectivity and less off-target activity [2,5], leading to less cardiotoxicity [7]. In 2019, ZAN was approved by the FDA as a second-line therapy for MCL [8]. In 2020, ORE was approved by the NMPA as a second-line therapy for CLL, SLL, and MCL [9].

As BTK inhibitors are used extensively in lifelong management regimens, toxicities are becoming problematic in the real-world setting. Off-target binding of the IBR, usually related to treatment-emergent adverse effects such as rashes, diarrhea, bleeding, infections, and atrial fibrillation, leads to treatment discontinuation in a substantial number (9–23%) of patients in clinical studies [10]. In comparison, ZAN and ORE have lower discontinuation rates. One study reported that adverse events leading to discontinuation were 7.8% in ZAN versus 13.0% in ibrutinib [11]. Another study on ORE reported 10% serious adverse events and a 6.4% discontinuation rate in the treatment of Waldenström’s macroglobulinemia [12].

Hematological malignancies make patients more susceptible to invasive fungal infections [13,14]. There is a risk of drug-drug interaction between triazole antifungals and the BTK inhibitors because ORE, ZAN, and IBR are metabolized mainly by CYP3A4 [2]. However, triazole antifungals such as ketoconazole are potent inhibitors of CYP3A4 [15]. This drug-drug interaction may increase the risk of the BTK inhibitors’ adverse reactions [11]. Experts have suggested that reducing the dose of IBR could be an attractive strategy to reduce toxicities [11]. However, this relationship between plasma exposure and toxicities has not been proven. Therefore, there is a necessity to develop an analytical method to study the relationship between plasma exposure and the clinical response of BTK inhibitors in patients, especially when combined with antifungals. To date, several articles have focused on the determination of IBR in human plasma or whole blood by LC-MS/MS [16,17,18,19,20,21]. Four studies [16,17,18,19] did not quantify the active metabolite, dihydroxydiol ibrutinib (DIH). The methods developed by Mukai et al. [20] and Koller et al. [21] involved a liquid/solid extraction step, which complicated the sample preparation procedure. Additionally, no article has reported the complete method of quantifying ORE and ZAN in human plasma using LC-MS/MS. In this study, we developed and validated a simple method with high sensitivity and specificity for the simultaneous determination of ORE, ZAN, IBR, and its active metabolite DIH in human plasma using LC-MS/MS. This method allows three kinds of clinical samples to be detected in one analytical batch, which is more convenient and faster.

## 2. Results and Discussion

### 2.1. Method Development

#### 2.1.1. Optimization of Mass Spectrometry Conditions

ORE, ZAN, IBR, DIH, and the IS are all characterized by alkaline functional groups so that the positive ESI mode can provide the greatest MS response with protonated molecular ions [M + H]^+^ as the precursor ion. The MRM transitions of *m*/*z* were selected as 428.1→411.2, 472.2→455.2, 441.1→304.2, 475.2→304.2, and 446.2→309.2 for the determination of the ORE, ZAN, IBR, DIH, and the IS, respectively. Then, the MS parameters were optimized, as shown in Table 1, to achieve the best signal intensities and reproducible fragmentation. The obtained product ion mass spectra and the structures of the analytes are shown in Figure 1.

#### 2.1.2. Optimization of Chromatography Conditions

The C18 column was considered suitable for separating the most moderately polar compounds. A ZORBAX Eclipse Plus C18 column (2.1 mm × 50 mm, 1.8 μm, Agilent, Santa Clara, CA, USA) was tried at first. The peak shapes of all analytes were found to be a bit trailing. Then, the ACQUITY UPLC HSS T3 column (2.1 × 50 mm, 1.8 μm, Waters, Milford, MA, USA), which is end-capped and beneficial to the peak shape of alkaline compounds, was used for the separation of the analytes in this study. The use of the chromatographic column showed acceptable retention times and peak shapes. Additionally, we tried 2 mM, 5 mM, and 10 mM ammonium formate in the water phase. The use of 10 mM ammonium formate not only maintained the stability of the mobile phase PH and ensured the reproducibility of the retention time but also provided the best peak shapes. Furthermore, 0.1% formic acid in the water phase enhanced the ionization of the analytes, which could increase the MS response. The flow rate of 0.5 mL/min also helped provide satisfactory peak shapes and suitable retention times. Finally, all analytes could be eluted in 4 min. After separation, the gradient elution also included a 1-min wash procedure (using 95% phase B) to elute other high-retention compounds in plasma and a 1.4-min re-equilibrate procedure to ensure that the elution of the next injection was not affected.

#### 2.1.3. Sample Preparation

In this method, simple protein precipitation was selected as the sample preparation procedure. Acetonitrile, which has a more robust precipitation capacity than methanol, was used to precipitate proteins from plasma. A plasma-to-acetonitrile ratio of one to four ensures complete protein precipitation. Considering that the sensitivity of DIH might not be enough, we evaporated the supernatant to dryness and redissolved it with a smaller amount of solution (approximately three-fifths of the supernatant) to improve the sensitivity of all analytes. In addition, using acetonitrile water (38:62, *v*/*v*) as the redissolved solution, which was in the same ratio as the initial mobile phase, could reduce the solvent effect and improve the peak shapes.

### 2.2. Method Validation

#### 2.2.1. Selectivity

Figure 2 shows the representative MRM chromatograms of blank plasma samples and samples at the lower limit of quantitation (LLOQ) and the upper limit of quantitation (ULOQ) level. There was no interference at the retention times of orelabrutinib (1.48 min), zanubrutinib (1.66 min), ibrutinib (3.45 min), dihydroxydiol ibrutinib (1.16 min), and ibrutinib-d5 (3.33 min), indicating that this method was highly selective and specific for detecting analytes.

#### 2.2.2. Matrix Effect and Recovery

The matrix effect and extraction recovery for the analytes in the plasma are summarized in Table 2. For the IS, the matrix effect was (98.1 ± 1.7)%, and the mean extraction recovery for analytes was found to be (92.0 ± 4.3)%. Therefore, this method had no significant matrix effects for all four analytes, which signified no ion suppression or enhancement in the plasma.

#### 2.2.3. Linearity, Lower Limit of Quantitation, Precision and Accuracy

The linear ranges, typical regression equations, and correlation coefficients of four analytes are shown in Table 3. The acceptance limit for linear curve fitting corresponds to the regression coefficient value (r^2^ > 0.99), which exhibited good linearity. The S/N of this method was greater than 10 when the LLOQ was 1.00 ng/mL for ORE, 0.400 ng/mL for IBR and DIH, and 2.00 ng/mL for ZAN, which indicated sufficient sensitivity. Five replicates of the QC samples, including LLQC, were analyzed in three separate runs to determine the accuracy and precision. Table 4 lists the intra- and inter-batch precision and accuracy for each sample. The intra- and inter-batch accuracy ranged from 94.7–105.7%, and the precision for both intra- and inter-batch were below 13%. These results demonstrated that precision and accuracy values were within the acceptance range.

#### 2.2.4. Carryover

Spiked samples at the ULOQ level and blank plasma samples were analyzed in rotation. The peak areas of the analytes and the IS in the blank samples were below 5% of those in the sample at the LLOQ level (3.9% for IBR, 2.9% for DIH, 1.4% for ORE, and 1.2% for ZAN), suggesting no significant carryover that affects the determination of subsequent samples.

#### 2.2.5. Stability

The data on the analytes’ stability in human plasma under different processing and storage conditions are summarized in Table 5. As shown in this table, the analytes were kept stable in plasma subjected to three freeze-thaw cycles from −40 °C or −80 °C to room temperature. The compounds could be stable in the autosampler at 10 °C for 24 h, indicating good post-extractive stability for the analytes. In addition, the analytes were kept stable for 6 h at room temperature, 2 h in ice baths and 30 days at −40 °C or −80 °C.

#### 2.2.6. Dilution Integrity

The mean deviations were −6.9%, −10.3%, −12.5, and −1.7% for ORE, ZAN, IBR, and DIH, respectively, and the precisions were below 4.1%. The results demonstrated that samples above the ULOQ but below the three-fold ULOQ could be accurately quantified after a five-fold dilution.

### 2.3. Application

In this study, a total of 80 clinical samples (*n* = 15 for IBR and DIH, *n* = 20 for ORE, and *n* = 45 for ZAN) were analyzed using the proposed method. The mean plasma concentrations at pre-dose (0 h) were (1.23 ± 0.83) ng/mL, (5.38 ± 3.83) ng/mL, (13.0 ± 11.0) ng/mL, and (12.4 ± 9.1) ng/mL, respectively, for IBR, DIH, ORE, and ZAN. The results showed significant individual differences between the plasma levels of IBR, DIH, ORE, or ZAN in patients with B-cell malignancies. This method was suitable for the analysis of these compounds in human plasma in the pharmacotherapy courses of IBR, ORE, and ZAN.

## 3. Materials and Methods

### 3.1. Chemicals and Reagents

The IBR (purity 98.0%), DIH (purity 98.0%), and IBR-d5 (internal standard, IS, purity 99.3%) were purchased from Toronto Research Chemicals Inc. (Toronto, ON, Canada). The ZAN is provided by BeiGene co. (Beijing, China). The ORE (purity 100.0%) was acquired from Nuocheng Jianhua Technology (Beijing, China). The acetonitrile and methanol of HPLC grade were obtained from Merck KGaA (Darmstadt, German). The dimethyl sulfoxide (DMSO) of HPLC grade was supplied by ROE Scientific Inc. (Newark, USA). The formic acid of HPLC grade was manufactured by Sigma-Aldrich (St. Louis, MO, USA). The ultrapure water was achieved by the Milli-Q system (Millipore, MA, USA). Additional reagents are analytically pure and are available on the market.

### 3.2. Analytical Equipment

The HPLC instrument was a 1290 series HPLC system (Agilent, Santa Clara, CA, USA), which comprised a pump (G4220A), an autosampler (G4226A), and a thermostatted column compartment (G1316C). An API 4000 triple quadrupole system (Sciex, Concord, ON, Canada) was employed as the mass spectrometric detector. The data were processed using the Analyst^®^ 1.6.2 software.

### 3.3. Chromatographic and Mass Spectrometric Conditions

The separation was performed using the ACQUITY UPLC HSS T3 column (2.1 × 50 mm, 1.8 μm, Waters, USA). The mobile phase comprised 10 mM ammonium formate containing 0.1 formic acid (Eluent A) and acetonitrile (Eluent B). For the duration of the chromatographic run, the flow rate was kept constant at 0.5 mL/min, while the column oven was maintained at 38 °C and the autosampler at 10 °C. The total run time was 6.5 min with a gradient elution program as follows: 0–4.0 min, B 38%; 4.0–4.1 min, B 38–95%; 4.1–5.0 min, B 95%; 5.0–5.1 min, B 95–38%; 5.1–6.5 min, B 38%. Furthermore, the sample volume for each injection is 3 µL.

Electrospray ionization (ESI), positive ion detection, and multiple reaction monitoring (MRM) scanning were employed for mass spectrometry data recording. Pure nitrogen was employed as the source gas. Optimized gas settings for the mass spectrometer included turbo gas of 65 pis, nebulizer gas of 55 pis, curtain gas of 40 pis, and collision gas of 10 pis, respectively. The electrospray ionization source was heated to 650 °C and operated in positive mode with an ion spray voltage of 5000 V. Table 1 lists the parameters of MRM transitions and compound-specific MS settings for every analyte.

### 3.4. Stock Solutions and Working Solutions Preparation

Independent stock solutions for each analyte were prepared at 0.500 mg/mL. The appropriate volumes of four stock solutions were mixed and then diluted by DMSO to provide a series of calibration working solutions. Similar steps were followed in the preparation of working solutions for quality control (QC) samples. The standard stock solution of IBR-d5 was dissolved in DMSO and then diluted using acetonitrile to obtain the IS working solutions with a final concentration of 20 ng/mL. The above solutions were stored at −40 °C.

### 3.5. Calibration Standards and QC Samples Preparation

Calibration standards and quality control samples were prepared by spiking plasma with the corresponding working solution in 1.5 mL tubes. Finally, the calibration standards contained 0.400 ng/mL to 200 ng/mL for IBR and DIH, 2.00 ng/mL to 1000 ng/mL for ZAN, and 1.00 ng/mL to 500 ng/mL for ORE. The QC samples were prepared at four concentration levels: low concentration quality control (LQC), middle concentration quality control (MQC), middle concentration quality control (GMQC), and high concentration quality control (HQC). The final concentrations were 1.00, 10.0, 40.0, and 160 ng/mL for IBR and DIH, 5.00, 50.0, 200, and 800 ng/mL for ZAN, and 2.50, 25.0, 100, and 400 ng/mL for ORE.

### 3.6. Plasma Sample Preparation

Initially, 200 µL of the IS working solution (containing IBR-d5 20 ng/mL) was added to 50 µL of plasma samples in 1.5 mL tubes. Then, the mixture was vortex mixed for 10 min and centrifugated at 16,000 rpm for 15 min. The supernatant was collected, evaporated to dryness, and then redissolved in 150 µL of acetonitrile water (38:62, *v*/*v*). Finally, the processed samples were transferred into autosampler vials for LC–MS/MS analysis.

### 3.7. Validation of the Method

The proposed assay was fully validated following the guidelines of the Food and Drug Administration for bioanalytical method validation [22]. The major validation parameters assessed were the LLOQ, linearity, precision, selectivity, matrix effect, recovery, carryover, and stability.

#### 3.7.1. Selectivity

For selective evaluation, blank plasma samples from six different lots and LLOQ samples were analyzed after pretreatment. Any interference peak eluted with either the analytes or the IS should be ≤20% of the LLOQ and ≤5% of the IS.

#### 3.7.2. Linearity and Lower Limit of Quantitation

Two replicates of the eight standard calibrators were prepared on consecutive days to evaluate the linearity. The standard curve in human plasma was generated by plotting the ratio of the peak area of the compounds to the IS against the standard concentration of the compounds. The weighted (1/x^2^) minimum linear regression was used to fit the linearity of the standard curves. The LLOQ was determined for a peak that gave a signal-to-noise (S/N) ratio of 10:1. The linear regression coefficient (r^2^) should be no less than 0.99. The accuracy of calibrators should be within ±15% of the concentration, except for the LLOQ (within ± 20%).

#### 3.7.3. Precision and Accuracy

Intra-batch and inter-batch precision and accuracy were estimated by analyzing five replicates for each concentration at LQC, MQC, GMQC, and HQC levels. The precision was measured as the relative standard deviation (RSD) between the nominal concentration and the actual measured concentration average. The accuracy was calculated as the relative error (RE) between the values. The acceptance criteria for two parameters were set at ±15%, except for the LLOQ, for which it should be within ±20%.

#### 3.7.4. Extraction Recovery and Matrix Effect

The QC samples at four concentration levels were analyzed to evaluate the extraction of the substance from the matrix during sample preparation. The mean extraction recoveries were calculated by comparing the peak area responses observed in pre- and post-spiked samples. The matrix effect was evaluated for the LQC and HQC samples. The matrix factor of analytes was calculated using the equation as follows: matrix factor (MF, %) = A/B × 100% (A is the peak area ratio of analytes added into the extracted plasma to IS; B is the peak area ratio of the analytes dissolved in the mobile phase to form IS). The precision of the MF calculated from the six lots of the matrixes should be ≤ 15%.

#### 3.7.5. Carryover

The carryover was determined by injecting a blank sample (processed surrogate matrix samples) immediately after running the highest calibration standard. The carryover was considered acceptable if the peak areas in the blank injection were more than 20 of the % LLOQ.

#### 3.7.6. Stability

The stability of all the analytes at low and high concentration levels was tested during both sample storage and handling procedures, including 6 h at room temperature, 3 freeze-thaw cycles from −40 °C and −80 °C to the room temperature, 30 days under −40 °C and −80 °C, 24 h in the autosampler (10 °C), and 2 h in ice baths. The stability can be considered valid when the RE and RSD do not exceed ± 15% and 15%, respectively.

#### 3.7.7. Dilution Integrity

The dilution quality control samples above the ULOQ at 200/200/1500/3000 ng/mL of DIH/IBR/ORE/ZAN were spiked, and a five-fold dilution using blank human plasma was applied before sample preparation. Five replicates were tested. Accuracy is required to be within 85% to 115% of the nominal concentrations, and precision should not exceed 15%.

### 3.8. Application

The method was used to analyze the plasma concentrations of ORE, ZAN, IBR, and its active metabolite DIH in clinical patients. Inpatients over 18 years old taking any of the above BTK inhibitors (ORE tablets, 150 mg, qd; ZAN tablets, 160 mg, bid; IBR tablets, 420 mg, qd) were eligible to participate in the study. Blood samples were taken pre-dose (0 h) and at 1.5 h, 2.0 h, 2.5 h, and 3.0 h post-dose. The blood samples were instantly transformed into heparinized vacuum tubes that contained K_2_EDTA after being drawn from the patients. Then, they were centrifugated at 4000 rpm and 4 °C for 8 min to obtain plasma samples. All clinical samples were stored at −80 °C until analysis. Ethical approval was obtained from the Ethics Committee of the First Affiliated Hospital of Nanjing Medical University. The research was conducted in accordance with the guidelines of the Declaration of Helsinki, and all patients signed the informed consent form.

## 4. Conclusions

In this study, a simple and specific LC-MS/MS method was developed for the simultaneous determination of ORE, ZAN, IBR, and its active metabolite DIH in human plasma. The run time was 6.5 min for each sample. The method was validated according to FDA guidelines. The method had good linearities between 1.00–500 ng/mL, 2.00–1000 ng/mL, and 0.400–200 ng/mL for ORE, ZAN, and IBR/DIH, respectively. This method exhibited good selectivity and recovery. The matrix effect was ignorable. Inter- and intra- run precision was below 13.0%. Inter- and intra-run accuracy was between −4.8% and 5.7%. Stability was confirmed under processing and storage conditions. Five-fold dilution integrity was validated. The method exhibited good applicability by quantifying the plasma levels of ORE, ZAN, IBR, and DIH in clinical patients.

## Figures and Tables

**Figure 1 molecules-28-01205-f001:**
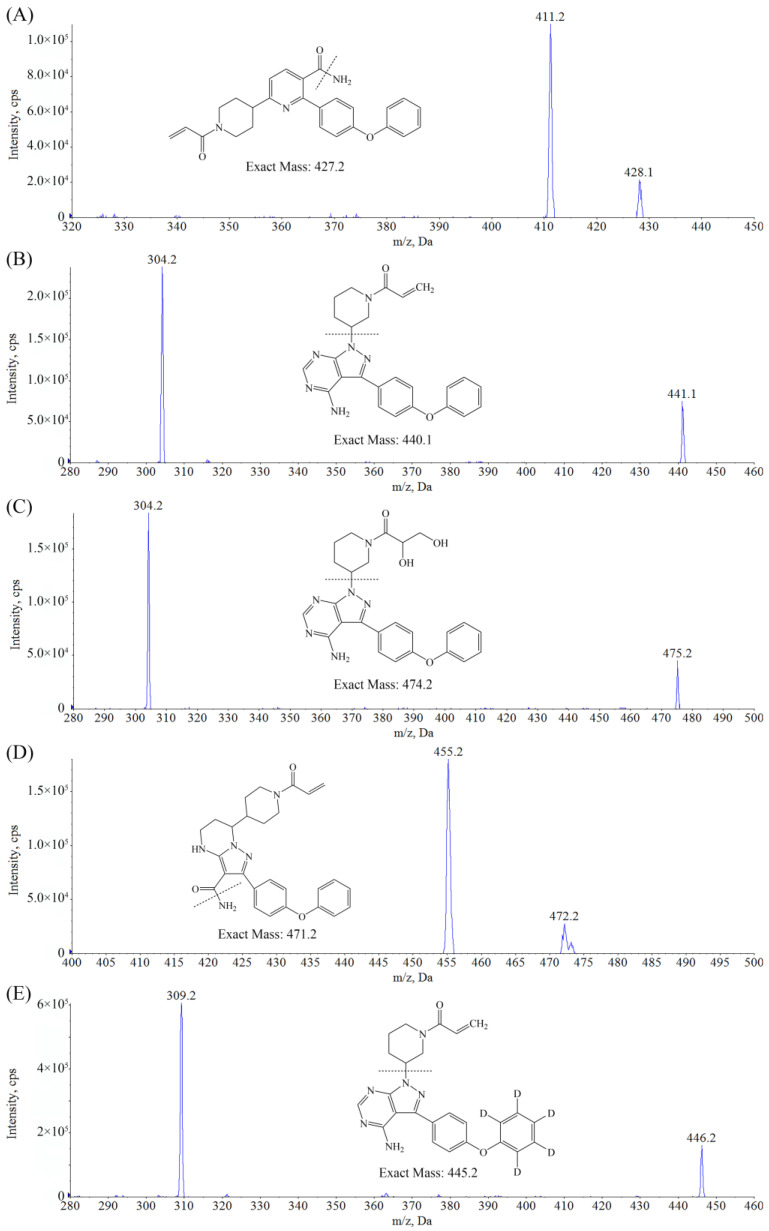
Product ion spectra of [M + H]^+^ and structures of (**A**) ORE, (**B**) IBR, (**C**) DIH, and (**D**) ZAN. (**E**) IBR-d5.

**Figure 2 molecules-28-01205-f002:**
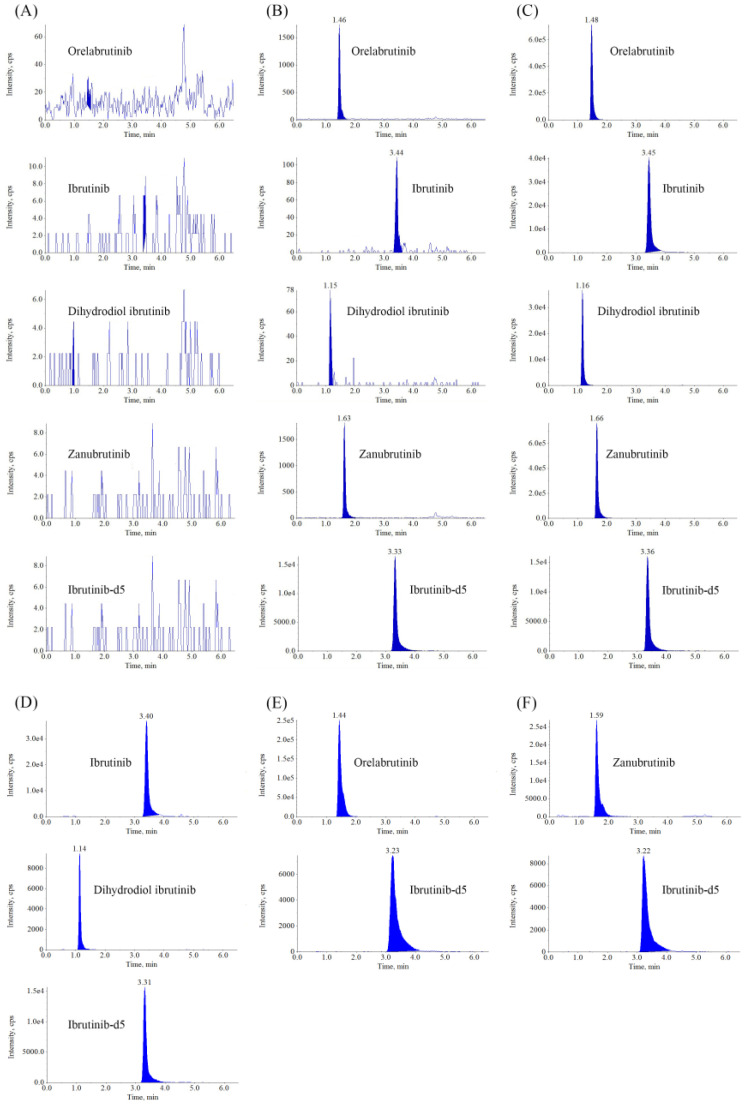
Representative MRM chromatograms of ORE, IBR, DIH, and ZAN in human plasma. (**A**) Blank plasma sample; (**B**) Plasma spiked with analytes at LLOQ and IS; (**C**) Plasma spiked with analytes at ULOQ and IS; (**D**) Plasma sample obtained from a patient at 1.5 h after oral administration of 420 mg IBR tablets; (**E**) Plasma sample obtained from a patient at 1.5 h after oral administration of 150 mg ORE tablets; (**F**) Plasma sample obtained from a patient at 1.5 h after oral administration of 160 mg ZAN tablets.

**Table 1 molecules-28-01205-t001:** MRM transitions and compound-specific MS settings.

Analytes	Precursor Ion (*m*/*z*)	Product Ion (*m*/*z*)	DP (V)	EP (V)	CE (V)	CXP (V)
IBR	441.2	304.2	109	7	40	15
DIH	475.2	304.2	108	9	43	15
ORE	428.1	411.2	104	8	29	15
ZAN	472.2	455.2	110	10	28	15
IBR-d5	446.2	309.2	109	7	40	15

DP, declustering potential; EP, entrance potential; CE, collision energy; CXP, collision cell exit potential.

**Table 2 molecules-28-01205-t002:** The results of the matrix effect and recovery.

Compound	Conc. Spiked (ng/mL)	Matrix Effect	Recovery
		Mean ± SD (%)	Mean ± SD (%)
IBR	1.00	102.7 ± 4.3	92.1 ± 4.7
	10.0	—	98.3 ± 3.4
	40.0	—	92.6 ± 2.6
	160	98.8 ± 2.2	92.8 ± 7.4
DIH	1.00	107.4 ± 4.6	99.2 ± 18.2
	10.0	—	94.9 ± 5.2
	40.0	—	96.4 ± 7.1
	160	106.8 ± 2.8	91.1 ± 5.3
ORE	1.00	100.5 ± 3.3	92.9 ± 3.8
	2.50	—	96.2 ± 2.3
	25.0	—	95.3 ± 5.3
	100	97.6 ± 2.7	94.8 ± 5.6
ZAN	400	99.5 ± 1.6	94.5 ± 4.4
	25.0	—	97.9 ± 2.8
	100	—	96.4 ± 5.6
	400	100.5 ± 2.7	94.8 ± 6.9

Conc., concentration; SD, standard deviation.

**Table 3 molecules-28-01205-t003:** Linear ranges, regression equations, and correlation coefficients of IBR, DIH, ORE and ZAN.

Analytes	Linear Range (ng/mL)	Regression Equation	Correlation Coefficient (r^2^)
IBR	0.400–200	*f* = 0.0119 × C + 0.000401	0.9986
DIH	0.400–200	*f* = 0.00611 × C + 0.0000178	0.9982
ORE	1.00–500	*f* = 0.0556 × C + 0.000915	0.9982
ZAN	2.00–1000	*f* = 0.0320 × C + 0.000852	0.9988

**Table 4 molecules-28-01205-t004:** Precision and accuracy of the LC-MS/MS method for the determination of IBR, DIH, ORE, and ZAN in human plasma (*n* = 3 runs with 5 replicates per run).

Analytes	Conc. Spiked (ng/mL)	Conc. Found (ng/mL)	Intra-Run		Inter-Run	
			RSD (%)	RE (%)	RSD (%)	RE (%)
IBR	0.400	0.401	12.2	0.2	13.0	1.7
	1.00	1.02	6.4	1.6	5.0	−1.2
	10.0	9.82	3.8	−1.8	3.0	−5.3
	40.0	39.7	2.5	−0.7	2.6	−1.0
	160	159	2.6	−0.2	2.2	−2.4
DIH	0.400	0.393	12.2	−1.4	8.6	−2.3
	1.00	1.01	6.9	1.2	6.5	−1.1
	10.0	10.0	3.6	0.3	3.4	0.0
	40.0	39.9	4.0	−0.2	4.1	−1.0
	160	158	5.1	−1.1	4.4	−4.8
ORE	1.00	0.987	5.3	−1.3	4.2	−1.9
	2.50	2.62	3.1	4.6	2.7	2.6
	25.0	25.6	3.6	2.3	3.0	−0.2
	100	102	2.8	1.9	2.8	2.0
	400	389	2.3	−2.7	2.3	−3.5
ZAN	2.00	1.96	4.5	−2.3	4.8	−2.1
	5.00	5.28	2.4	5.7	2.2	4.2
	50.0	50.9	2.9	1.3	3.0	0.1
	200	202	2.3	0.8	2.3	1.4
	800	773	2.8	−3.4	2.8	−4.6

Conc., concentration; RSD, relative standard deviation; RE, relative error.

**Table 5 molecules-28-01205-t005:** Stability data of analytes under different storage conditions (*n* = 3).

Stability	Storage Condition	IBR			DIH			ORE			ZAN		
		Group (ng/mL)	RSD (%)	RE (%)	Group (ng/mL)	RSD (%)	RE (%)	Group (ng/mL)	RSD (%)	RE (%)	Group (ng/mL)	RSD (%)	RE (%)
Autosampler stability	(10 °C, 24 h)	1.00 160	8.0 2.4	−4.5 −1.7	1.00 160	12.1 5.5	1.0 −3.3	2.50 400	3.5 3.3	4.7 −7.5	5.00 800	2.3 2.0	0.1 −6.9
Ice bath stability	(0 °C, 2 h)	1.00 160	4.1 1.7	−5.8 −1.9	1.00 160	11.2 3.3	1.5 −2.7	2.50 400	7.6 2.6	4.1 −3.2	5.00 800	5.0 3.1	5.5 1.5
Benchtop stability	(20 °C, 6 h)	1.00 160	5.3 4.0	−5.8 −1.3	1.00 160	2.8 2.8	8.7 6.0	2.50 400	5.4 3.6	−2.8 1.1	5.00 800	1.5 4.2	1.8 0.6
Freeze-thaw stability	(−40 °C, 3 cycles)	1.00 160	6.5 3.7	−0.9 −2.9	1.00 160	4.1 4.8	4.2 4.4	2.50 400	6.8 6.6	3.9 −3.0	5.00 800	6.9 4.9	−1.4 −2.3
Freeze-thaw stability	(−80 °C, 3 cycles)	1.00 160	7.3 4.4	1.6 0.0	1.00 160	3.0 6.7	7.7 −1.0	2.50 400	5.8 1.7	5.7 −5.8	5.00 800	5.6 4.5	1.5 −2.5
Long-term	(−40 °C, 30 d)	1.00	−8.5	3.9	1.00	8.0	3.3	2.50	1.1	7.9	5.00	5.2	4.7
stability		160	−10.2	2.6	160	0.6	1.1	400	−7.7	6.0	800	−8.1	5.1
Long-term stability	(−80 °C, 30 d)	1.00	−3.4	5.0	1.00	0.0	8.0	2.50	6.1	4.7	5.00	7.5	7.7
		160	−7.3	2.7	160	−1.0	4.3	400	−2.0	5.1	800	−3.9	5.5

## Data Availability

The data presented in this study are available upon request from the corresponding author.

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
