# Peer review of "Simultaneous Determination of Orelabrutinib, Zanubrutinib, Ibrutinib and Its Active Metabolite in Human Plasma Using LC-MS/MS"

_molecules, 2023, doi:10.3390/molecules28031205_

Round 1
Reviewer 1 Report
1. The first sentence of abstract is irrelevant to the following content. What do you want to show?
2. Optimization process of defect pre-processing.
3. The full text lacks innovation.
Author Response
Response to Reviewer 1 Comments
Point 1: The first sentence of abstract is irrelevant to the following content. What do you want to show?
Response 1: Thanks for pointing out this. We intended to introduce the effects of the drugs to be analyzed. In the revised manuscript, ‘Bruton’s tyrosine kinase inhibitors have greatly improved the treatment of B-cell malignancies’ has been changed to ‘Ibrutinib, orelabrutinib and zanubrutinb are all Bruton’s tyrosine kinase inhibitors, which have greatly improved the treatment of B-cell malignancies.’
Point 2: Optimization process of defect pre-processing.
Response 2: Thanks for pointing out this. Simple protein precipitation can satisfy the need for method development. In the revised manuscript, section 2.1.3. Sample preparation’ has been added to describe our considerations when developing the sample preparation approach as follows:
2.1.3. Sample preparation
In this method, protein precipitation was selected as the sample preparation procedure. Acetonitrile, which has a more robust precipitation capacity than methanol, was used to precipitate proteins from plasma. A plasma-to-acetonitrile ratio of 1 to 4 ensures complete protein precipitation. Considering that the sensitivity of dihydroxydiol ibrutinib might not be enough, we evaporated the supernatant to dryness and redissolved with a smaller amount of solution (about three-fifths of the supernatant) to improve the sensitivity of all analytes. In addition, acetonitrile-water (38:62, v/v) as the redissolved solution, which was in the same ratio as the initial mobile phase, could reduce the solvent effect and improve the peak shapes.
Point 3: The full text lacks innovation.
Response 3: Thanks for your comments. As outlined in the revised introduction, the principal reason for BTK inhibitors’ failure is intolerance rather than disease progression (adverse effects mainly caused discontinuation). The drug-drug interaction between triazole antifungals and BTK inhibitors is a common clinical condition, so exposure to BTK inhibitors, when combined with antifungal drugs, needs to be studied. Additionally, this drug-drug interaction has been reported may increase the risk of BTK inhibitors’ adverse reactions due to their increased plasma exposure, but there have been no definitive conclusions. So, the relationship between plasma exposure and the clinical response of BTK inhibitors in patients remains to be studied. Both points require a reliable method for the determination of BTK inhibitors in human plasma. In this study, we intended to develop and validate an LC-MS/MS method suitable for quantifying three BTK inhibitors and the metabolite. It is known that no article has reported the complete method of the determination of orelabrutinib and zanubrutinib in human plasma. Compared with the two available articles reporting the determination of ibrutinib and dihydroxydiol ibrutinib, our method was simpler and could be more suitable for analyzing large quantities of clinical samples. In addition, we usually receive clinical samples containing different drugs (different patients take different BTK inhibitors) at the same time, and our method allows all clinical samples to be detected in one analytical batch, which is more convenient and faster.
In the revised manuscript, section introduction, content about the side effects of BTK inhibitors has been added as follows:
As BTK inhibitor is used extensively in lifelong management regimens, toxicities are becoming problematic in the real-world setting. Off-target binding of IBR, usually related to treatment-emergent adverse effects such as rash, diarrhea, bleeding, infections and atrial fibrillation, leads to treatment discontinuation in a substantial number (9-23%) of patients in clinical studies [10]. In comparison, ZAN and ORE have lower discontinuation rates. One study reported that adverse events leading to discontinuation were 7.8% in ZAN versus 13.0% in ibrutinib [11]. Another study on ORE reported 10% serious adverse events and a 6.4% discontinuation rate in the treatment of Waldenström's macroglobulinemia [12].
[10] Estupiñán, H.Y.; Berglöf, A.; Zain, R.; Smith, C.I.E. Comparative Analysis of BTK Inhibitors and Mechanisms Underlying Adverse Effects. Front Cell Dev Biol. 2021, 9, 630942, doi:10.3389/fcell.2021.630942.
[11] Cho, H.J.; Baek, D.W.; Kim, J.; Lee, J.M.; Moon, J.H.; Sohn, S.K. Keeping a balance in chronic lymphocytic leukemia (CLL) patients taking ibrutinib: ibrutinib-associated adverse events and their management based on drug interactions. Expert Rev Hematol 2021, 14, 819-830, doi:10.1080 /17474086.2021.1967139.
[12] Cao, X.X.; Jin, J.; Fu, C.C.; Yi, S.H.; Zhao, W.L.; Sun, Z.M.; Yang, W.; Li, D.J.; Cui, G.H.; Hu J.D.; et al. Evaluation of orelabrutinib monotherapy in patients with relapsed or refractory Waldenström's macroglobulinemia in a single-arm, multicenter, open-label, phase 2 study. EClinicalMedicine 2022, 52, 101682, doi:10.1016/j.eclinm.2022.101682.
Additionally, ‘It has been reported that the drug-drug interaction between ibrutinib and triazole antifungals may increase the toxicities of ibrutinib due to its increase of plasma exposure [13]. There is a necessity to study the relation between plasma exposure and clinical response of BTK inhibitors in patients, especially when combined with antifungals.’ has been changed to ‘Experts have suggested that reducing the dose of IBR could be an attractive strategy to reduce toxicities [11]. However, this relationship between plasma exposure and toxicities have not been proven. Therefore, there is a necessity to develop an analytical method to study the relationship between plasma exposure and the clinical response of BTK inhibitors in patients, especially when combined with antifungals.’
Reviewer 2 Report
Simultaneous determination of orelabrutinib, zanubrutinib, ibrutinib and its active metabolite using LC-MS/MS
Overall comments:
This study was performed to optimize the method process of simultaneous analysis of 4 compounds, orelabrutinib, zanubrutinib, ibrutinib and dihydrodiol ibrutinib in human plasma by LC/MSMS. This pharmaceutics are used to treat the B-cell malignancies. The analysis method is optimized for each individual condition including: mass spectrometry conditions, chromatography conditions, then evaluated through selectivity, matrix effect and recovery, linearity, LOQ, precision and accuracy, stability. Even standard and blank samples were used to conduct the study. The analytical method was then applied to 80 samples and showed good applicable results.
The study found itself to be novel in the simultaneous analysis of 4 compounds that had not been performed in previous studies. On the other hand, the method of sample treatment is also simpler to avoid loss of analyte.
General discussion:
1. The article is novel in analyzing 4 compounds at the same time. But as far as I know, these compounds are drugs used to treat different types of malignancies. Therefore, each cancer patient will use a different drug, so it is not necessary to develop a simultaneous method to analyze these 4 compounds, because each patient can use separately a different analytical method. This was shown in the application results, with no sample detecting all four compounds.
2. It is necessary to clarify the significance of the determination of these substances in human plasma. Is it because these drugs are used to treat diseases that their residues will be remained in the plasma? What does this identification mean in treating the patient?
3. The research is done quite simply, but the manuscript presented is difficult to understand, without much scientific significance. The development of a method for the simultaneous analysis of these substances is only looked a small technical improvement with only four substances.
Specific comments:
1. Title: Need to add “in human plasma” in the title
2. Introduction section should be introduced about each compound and its effects. On the other hand, it is also necessary to present what the meaning of this study in the introduction section.
3. Formatting and numbering are very messy. The order of the large items is wrong, it should be adjusted as follows: 1. Introduction (correct, page 1); 2. Materials and methods (incorrect, page 7); 3. Results and discussion (incorrect, page 2).
4. The study was performed with pretreatment with protein precipitation with acetonnitrile but no recovery of this pretreatment was reported in the results.
5. The study used internal standard ibrutinib-d5 but the manuscript did not present which step of the analysis method was added and what was it used for?
6. Four substances should be abbreviated. What is ULOQ stand for? Should use “LOQ” instead of “LLOQ”.
7. Eluent A, should be presented as ultrapure water+10 mM ammonium formate containing 0.1 formic acid which can be considered as buffer. Please explain why the ammonium formate and the formic acid were not added in eluent B. Because when B reaches 95%, the amount of buffer in the chromatographic column is very low.
8. The result sections are on optimization but only shows the final results of the optimal values. Therefore, it is not possible to understand how to achieve those optimal conditions, especially for the optimization of chromatography conditions section. No description of chromatographic flow rates is found in this section.
9. Need to show some example chromatograms for positive clinical samples
Conclusions:
Overall, the study was quite simple with the simultaneous analysis of only 4 compounds, just developing a minor analytical technique and not an analytical method development, although validation was also carried out according to the FDA guideline. The article is poorly written and confusing. The article needs a lot of editing in English.
Round 2
Reviewer 1 Report
All the questions were answered very well, and the manuscript can be accepted.
Reviewer 2 Report
The manuscript has been revised according to reviewer's request.